# Identification of Loci Associated with Enhanced Virulence in Spodoptera litura Nucleopolyhedrovirus Isolates Using Deep Sequencing

**DOI:** 10.3390/v11090872

**Published:** 2019-09-17

**Authors:** Mark P. Zwart, Ghulam Ali, Elisabeth A. van Strien, Elio G. W. M. Schijlen, Manli Wang, Wopke van der Werf, Just M. Vlak

**Affiliations:** 1Department of Microbial Ecology, The Netherlands Institute of Ecology (NIOO-KNAW), 6708PB Wageningen, The Netherlands; 2Laboratory of Virology, Wageningen University and Research, 6708PD Wageningen, The Netherlands; 3Bioscience, Wageningen Plant Research, Wageningen University and Research, 6708PD Wageningen, The Netherlands; 4Wuhan Institute of Virology, Chinese Academy of Sciences, Wuhan 430071, China; 5Centre for Crop Systems Analysis, Wageningen University and Research, 6708PD Wageningen, The Netherlands

**Keywords:** Spodoptera litura nucleopolyhedrovirus, SpltNPV, virulence, deep sequencing, Illumina, genomics

## Abstract

*Spodoptera litura* is an emerging pest insect in cotton and arable crops in Central Asia. To explore the possibility of using baculoviruses as biological control agents instead of chemical pesticides, in a previous study we characterized a number of S. litura nucleopolyhedrovirus (SpltNPV) isolates from Pakistan. We found significant differences in speed of kill, an important property of a biological control agent. Here we set out to understand the genetic basis of these differences in speed of kill, by comparing the genome of the fast-killing SpltNPV-Pak-TAX1 isolate with that of the slow-killing SpltNPV-Pak-BNG isolate. These two isolates and the SpltNPV-G2 reference strain from China were deep sequenced with Illumina. As expected, the two Pakistani isolates were closely related with >99% sequence identity, whereas the Chinese isolate was more distantly related. We identified two loci that may be associated with the fast action of the SpltNPV-Pak-TAX1 isolate. First, an analysis of rates of synonymous and non-synonymous mutations identified neutral to positive selection on open reading frame (ORF) 122, encoding a viral fibroblast growth factor (vFGF) that is known to affect virulence in other baculoviruses. Second, the homologous repeat region hr17, a putative enhancer of transcription and origin of replication, is absent in SpltNPV-Pak-TAX1 suggesting it may also affect virulence. Additionally, we found there is little genetic variation within both Pakistani isolates, and we identified four genes under positive selection in both isolates that may have played a role in adaptation of SpltNPV to conditions in Central Asia. Our results contribute to the understanding of the enhanced activity of SpltNPV-Pak-TAX1, and may help to select better SpltNPV isolates for the control of *S. litura* in Pakistan and elsewhere.

## 1. Introduction

The leafworm *Spodoptera litura* is an emerging pest insect in Southeast Asia and the Indian subcontinent [1], more specifically in cotton and arable crops in Pakistan. Control of this insect is predominantly through the use of chemical insecticides. However, this strategy is unsustainable in the long run due to resistance developing in the insect as well as the negative impact of chemical insecticides on the health of humans and the environment. Safe, effective, and sustainable alternatives therefore are being sought for control of this insect. Baculoviruses comply with these characteristics for alternatives [2,3] and are therefore being investigated for control of this insect in Pakistan.

In a field survey in 2011–2013, 22 baculovirus isolates were obtained from *S. litura* from different cropping systems [4,5]. These isolates turned out to be nucleopolyhedroviruses (NPVs), members of the genus *Alphabaculovirus* within the *Baculoviridae* family [6,7]. The high sequence identity (>98%) suggests that these isolates belong to the *Spodoptera litura nucleopolyhedrovirus* (SpltNPV) species [8]. These isolates also form a cluster of genotypic variants that are closely related to each other and are distinct from the SpltNPV-type species (SpltNPV-G2, NC_003102.1) [9]. Within these 22 Pakistan isolates, three genogroups were identified by restriction fragment length polymorphism (RFLP) analysis [4,5]. These genogroups strongly correlate to the regions where they were found and to a much lesser extent to the cropping system [5]. It was also found that four SpltNPV isolates, all originating from the Punjab, showed a faster speed-of-kill than the other SpltNPVs [5,10]. The close relatedness among the Pakistani SpltNPV isolates and their more distant relatedness to the SpltNPV-type species from China and SpltNPV isolates from Japan suggests that SpltNPV may have recently been introduced to Pakistan. The Pakistan SpltNPVs may be in an early stage of adaptive radiation, providing a unique opportunity to study this epidemiological, ecogeographical, and evolutionary process.

Next-generation sequencing (NGS) is the very high-throughput sequencing of the genetic material of an organism or community, using massively parallel approaches and typically employing a shotgun approach to library preparation. Due to the volume of sequencing data generated, the introduction of NGS has also led to the rapid development of bioinformatic tools to analyze large sequencing datasets. The decrease in time and cost of sequencing samples have brought about a revolution in virology [11,12,13,14], whilst the technology is continuously being improved [11]. An equally important advantage of NGS is the possibility of identifying rare polymorphisms in mixed-genotype virus populations by increasing the coverage with which they are sequenced. For example, traditional virological approaches such as restriction fragment length polymorphism (RFLP) analysis on clones has allowed for the identification of many different genotypes within baculovirus populations [15,16,17]. Ultra-deep sequencing can be used to identify systematically and quantitatively genetic variation that is present at very low frequencies [18].

In this paper we characterize the full-genomes of the SpltNPV-Pak-BNG and SpltNPV-Pak-TAX1 isolates, with low and high speed-of-kill respectively [5,10], using an NGS approach. RFLP analysis showed that these two isolates belong to different genogroups, with SpltNPV-Pak-BNG belonging to genogroup A and SpltNPV-Pak-TAX1 belonging to genogroup B [5]. We therefore also seek to further detail the differences between the SpltNPV genogroups A and B, by studying representative isolates for each group. When whole genome sequences are considered, the question is whether these isolates share 99% nucleotide identity, as suggested by the sequencing of a limited number of open reading frames [4,5]. In addition, at what loci on the SpltNPV genome do genetic differences occur? Second, how do these viruses from Pakistan compare with strains from China? Third, we sought to better understand the genetic basis of the difference in speed of kill of the above two isolates. Can we identify loci associated with the higher virulence of the SpltNPV-Pak-TAX1 isolate? Finally, we are interested in determining how much genetic diversity there is within each isolate in view of the supposed recent occurrence in a new region (Pakistan). Do the Pakistani SpltNPV isolates contain considerable genetic variation, as has been found for many other baculoviruses, or not? We found that the two Pakistani isolates were indeed closely related, being more similar to each other than to a Chinese reference strain. We identified two loci that are likely to be associated with the higher virulence of SpltNPV-Pak-TAX1. We also found very little genetic diversity within these isolates.

## 2. Materials and Methods

### 2.1. SpltNPV Isolates, Insect Rearing, Virus Amplification, and DNA Extraction

The SpltNPV isolates SpltNPV-Pak-BNG and SpltNPV-Pak-TAX1 were obtained from Bahawalnagar and Taxila (Punjab, Pakistan) falling into genogroups A and B, respectively [5]. The viruses were propagated in third instar *S. litura* larvae at the National Agricultural Research Center in Islamabad, Pakistan as described in [4]. Occlusion bodies (OBs) from the Chinese isolate SpltNPV-G2b were obtained from Professor Kai Yang from Sun Yat-sen University. We have named this virus isolate SpltNPV-G2b because it is derived from the previously analyzed Splt-G2 [9]. The DNA was extracted from occlusion-derived virions (ODVs), which were in turn were isolated from occlusion bodies (OBs) as described in [4].

### 2.2. Library Preparation and Deep Sequencing

All libraries for deep sequencing were prepared using the Nextera XT kit (Illumina, San Diego, CA, USA), following the manufacturer’s instructions and using recommended adapters. For SpltNPV-Pak-BNG and SpltNPV-Pak-TAX1, libraries were then sequenced by paired-end Illumina MiSeq. Library preparation and sequencing were performed at the Bioscience unit of Wageningen University and Research (Wageningen, The Netherlands). For SpltNPV-G2b, the library was sequenced by Illumina HiSeq X Ten 150 paired-end. Library preparation and sequencing were performed at Beijing Novogene Bioinformatics Technology Co., Ltd (Beijing, China).

### 2.3. Analysis of Deep Sequencing Data

For primary analyses of the deep sequencing data, we used CLC Genomics Workbench v11.0.1 (Qiagen Genomics, Aarhus, Denmark). We used two different approaches to analyze the data: (1) a “resequencing” approach, in which reads are mapped to an existing reference genome, and (2) a de novo assembly approach, in which the reads were assembled without the help of a reference genome.

For analyzing the data with a resequencing approach, the demultiplexed reads were trimmed using the trim sequences tool with default settings, except that the minimum Phred score was raised to 30 (and the expected error rate lowered to ≤ 1/1000) because we expected to have high coverage. Next, the trimmed reads were mapped to the reference genome (NC_003102.1 or SpltNPV-G2b), low frequency variant detection was carried out, and consensus sequences were exported using default settings. For further analysis of the data, we then exported a detailed mapping report (*.tsv) and the low frequency variant detection tables to R v3.4.3 [19].

Analyzing the data with a de novo assembly approach was only feasible for the MiSeq data obtained for SpltNPV-Pak-BNG and SpltNPV-Pak-TAX1. Reads were trimmed as for the resequencing approach. Subsequently a de novo assembly was run using the default CLC Genomics Workbench settings, ignoring contigs shorter than 1000 bp. The consensus sequence of the contig corresponding to the full-length virus genome (see Results) was then extracted. We set the start codon of the *polyhedrin* gene (open reading frame (ORF) 1) as the first reference position in the circular genome. We then used these consensus sequences for two subsequent analysis. First, to identify differences between the two isolates, we performed a reciprocal mapping of reads (i.e., BNG reads on the TAX1 de novo assembly consensus sequence, and vice versa), followed by low frequency variant detection. Second, to identify within-isolate polymorphism we mapped the reads of an isolate to its own de-novo-assembled consensus sequence.

### 2.4. Identification of Mutations

We classified mutations as being small or large, based on the methods used to detect them. Small mutations are short enough to be identified using individual reads by the low frequency variant detector in CLC Genomics Workbench, and in this case included single nucleotide polymorphisms (SNPs), multinucleotide polymorphisms (MNPs) and short indels (<100 bp). The high-frequency variant detector does not report indels that span over regions longer than the typical read length, and so we had to resort to other methods. The method used to detect large indels was different for the resequencing and de novo assembly approaches.

To map large indels in the resequencing analysis, we scanned for low coverage regions (<10 reads) in the mapping of the reads to the reference genome. The coverage in these low coverage regions was essentially zero. We then searched for the consensus sequence of the non-mapping portion of reads next to the low coverage area in the genome. In most cases found this corresponded to the sequence adjacent to the opposite end of the low coverage region. Similarly, we checked where the split-mapped reads were mapping, and this provided further support for the occurrence of deletions. (i.e., pair-end reads can map on both sides of the deletion, and because of the distance covered on the reference genome can be considered split-mapped reads.). To map large indels for the de novo assembly approach, we aligned the consensus de novo assembly sequence to the reference genome, and then checked for gaps.

### 2.5. dN/dS and dI/dS Analyses

The ratio of the rate of nonsynonymous to the rate of synonymous substitution (dN/dS) is widely used to identify the kinds of selection acting on genomes or genes. The main assumption behind this analysis is that nonsynonymous substitutions can be advantageous or disadvantageous because they lead to changes in the amino acid sequence, whereas synonymous changes are neutral because they do not alter the amino acid sequence. When dN/dS < 1 there is purifying selection (more synonymous mutations then expected by chance), whereas when dN/dS > 1 there is directional selection (more nonsynonymous mutations then expected by chance). When dN/dS = 1, there is neutral evolution, as with mutations that are assumed to be neutral or not occur at the same rate. Following [20], we also analyzed the rate of evolution in intergenic sequences (dI/dS), using the rate of synonymous mutations (dS) in coding regions to establish a baseline for the rate of mutations in intergenic regions (dI).

To perform a dN/dS analysis on the whole genome and individual genes, we followed an approach similar to that used by [20]. First we generated an expectation for the number of possible synonymous (S) mutations within open reading frames (ORFs) that do not lead to a change in the amino acid sequence and nonsynonymous (N) mutations within ORFs that do lead to changes in the amino acid sequence, using the SpltNPV-G2b as a reference sequence. We tallied the number of times each codon was used in the whole genome or in a gene, and multiplied this number by the synonymous or nonsynonymous mutations possible for that particular codon. The sum of these products is then the numerator for each rate. The synonymous or nonsynonymous mutations for each codon were weighted by the probability of the underlying nucleotide substitution. We are not aware of any mutational spectrum data that could be used to infer the mutational biases for an *Alphabaculovirus* and we therefore simply assumed a transition to transversion ratio (TTR) of either 1, 3, or 5, and generated predictions for each value. Custom R scripts were used to generate these predictions. Finally, the numerator for each rate was obtained by counting the number of fixed (i.e., not polymorphic) synonymous or nonsynonymous substitutions observed in each gene, or over the whole genome. Some genes contained non-synonymous mutations but no synonymous mutations. To be able to approximate dS in these cases, we calculated the expected number of synonymous mutations in the gene, given the genome-wide occurrence of synonymous mutations and the length of the gene.

In the dI/dS analysis, the denominator (dS) is exactly the same as in the dN/dS analysis. The rate of intergenic mutations (dI) was determined by first generating a prediction for the number of possible intergenic substitutions, by considering the frequency of all nucleotides in intergenic regions (i.e., all positions outside of ORFs). All possible mutations were also weighted for mutational bias, assuming different TTR values. Finally the numerator, dI, was obtained by counting all intergenic mutations. All R code used in the manuscript is available in the Appendix A.

## 3. Results

### 3.1. Generation of SpltNPV-G2b Reference Genome

We set out to better understand the high virulence of the Splt-Pak-TAX1 isolate by comparing its genome to that of a lower virulence isolate, Splt-Pak-BNG. To aid our bioinformatic analysis and to compare Pakistani SpltNPV isolates to an endemic Chinese SpltNPV strain, we Illumina sequenced a Chinese SpltNPV isolate (henceforth SpltNPV-G2b) which is derived from the type isolate (SpltNPV-G2, NC_003102.1) [9]. With the relatively short Illumina HiSeq reads, we could not de novo assemble a complete genome, and therefore used a resequencing approach using SpltNPV-G2 as a reference sequence, resulting in high level of coverage (resequencing mean coverage ± SD: 8784 ± 1388). The consensus SpltNPV-G2b sequence is highly similar to SpltNPV-G2, sharing 99.77% identity at the nucleotide level and being of similar length (SpltNPV-G2: 139,342 bp, SpltNPV-G2b: 139,327). There are only 320 differences between SpltNPV-G2 and SpltNPV-G2b, of which 47 are short gaps, and no evidence of large genomic deletions, insertions, or rearrangements was found. An overview of sequencing coverage is given in Appendix A, and the SpltNPV-G2b sequence is in GenBank (MN342245). We used the SpltNPV-G2b sequence as a reference for all subsequent reference-based analyses. There were no major differences in terms of gene annotation between the SpltNPV-G2 and SpltNPV-G2b sequences, and therefore ORFs for these two isolates refer to the same gene. ORF numbers reported in the results section always refer to SpltNPV-G2/G2b for convenience.

### 3.2. Overview of Sequencing Results for SpltNPV-Pak-BNG and SpltNPV-Pak-TAX1

We analyzed the SpltNPV-Pak-BNG and SpltNPV-Pak-TAX1 Illumina data through two different approaches: a resequencing approach that employs the SpltNPV-G2b as a reference and a de novo assembly approach. As we deep-sequenced purified virus DNA, despite using only high quality reads (Phred ≥ 30), we still had high coverage of the viral genome (resequencing mean coverage ± SD: SpltNPV-Pak-BNG: 4545 ± 1060; SpltNPV-Pak-TAX1: 5746 ± 1187). For the resequencing analysis, for both isolates there are approximately 2000 bases in the reference genome without any coverage (Appendix A). Likewise, for the de novo assembled genomes, the total genome length for the two Pakistani isolates (SpltNPV-Pak-BNG: 137,155 bp, SpltNPV-Pak-TAX1: 137,655 bp) is shorter than the reference genome SpltNPV-G2b (139,327 bp). Therefore, both approaches suggest there are large indels present in both of the Pakistani genotypes as compared to the reference genome SpltNPV-G2b.

To gauge whether further genomic variation identified by both methods was similar, we compared the number of fixed SNPs unique to SpltNPV-Pak-BNG and SpltNPV-Pak-TAX1 identified by each method. The total number SNPs was similar for the two methods, although the de novo assembly called fewer mutations (resequencing 482, de novo assembly: 445). The total number of SNPs found in each ORF was similar for the two approaches (Spearman rank correlation: ρ = 0.984, *n* = 242, *p* < 0.001). There does appear to be a conflict for ORF4 and ORF125 (Appendix A), and in both cases the resequencing approach identifies more mutations than the de novo assembly. For ORF125, this discrepancy between the two approaches can be explained by the occurrence of a large indel at this locus for the Pakistani SpltNPVs (See Results 3.3). For ORF4, the *hoar* gene, there are discrepancies with the original SpltNPV-G2 sequence [9], probably related to repeat regions, which lead to mapping errors for the resequencing approach. As a final validation of our deep sequencing analyses, we compared the *Eco*RI restriction fragment length profiles (RFLP) predicted by the consensus sequences for the de novo assemblies of the two viruses with actual *Eco*RI digests [5]. The predicted and observed RFLP profiles were highly similar (Appendix A), providing further confirmation of the accuracy of the deep sequencing analysis.

The SpltNPV-Pak-BNG and SpltNPV-Pak-TAX1 genomes have been annotated (Appendix A) and compared to the reference strain SpltNPV-G2 [9] and to the genome of a closely related virus, Spodoptera littoralis nucleopolyhedrovirus (SpliNPV) [21]. In addition to the loss of a limited number of ORFs due to the large deletions (Section 3.3), SpltNPV-Pak-TAX1 is missing ORF9. The latter is a very small ORF and may be not functional. However, overall the organization of the SpltNPV-Pak-BNG and SpltNPV-Pak-TAX1 genomes is highly similar to SpltNPV-G2.

### 3.3. Occurrence of Large Indels in both Virus Isolates and the Deletion of hr17 in SpltNPV-Pak-TAX1

Both sequencing methods suggest that there are indels in both SpltNPV-Pak-BNG and SpltNPV-Pak-TAX1, both with respect to each other and to the reference sequence SpltNPV-G2b. For the resequencing data, plotting the coverage per position shows that low coverage positions cluster, and that there are two large deletions with respect to the reference sequence in each Pakistani isolate (Appendix A). The exact positions of these indels were determined using both sequence analysis methods (see Materials and Methods section), and were found to be identical (Table 1). For both isolates, the largest indel results in the loss of the entire ORF126. In SpltNPV-Pak-BNG, another indel partially removes ORF125, whereas in SpltNPV-Pak-TAX1 a second indel entirely removes hr17, one of 17 homologous regions found in the SpltNPV reference sequence and isolate BNG. The (putative) functions of ORF125 and ORF126 are unknown.

### 3.4. High Similarity between the Two Pakistani Virus Isolates

We were only able to obtain de novo assembled full-length genomes for SpltNPV-Pak-BNG and Splt-Pak-Tax1, whereas the resequencing approach could be used for all three SpltNPV virus isolates studied here. When we compare the three consensus genome sequences obtained by resequencing with six different metrics, we find that all three virus isolates are closely related (Figure 1). However, SpltNPV-Pak-BNG and SpltNPV-Pak-TAX1 are more similar to each other than to SpltNPV-G2b for five out of six metrics. This result is congruous with previous work, in which RFLPs and Sanger sequencing of a limited number of ORFs also suggested the two isolates were very similar [4,5]. We speculate that these results suggest that SpltNPV-Pak-BNG and SpltNPV-Pak-TAX1 diverged from a single introduction of SpltNPV into Pakistan.

For our subsequent detailed analysis of mutations, we decided to focus on two classes of mutations. First, we consider mutations with respect to SpltNPV-G2b that are shared by the two isolates from Pakistan. These mutations can render information on the longer-term evolution of SpltNPV in Pakistan, but are not likely to be important for differences in virulence between the virus isolates from Pakistan. This comparison can be performed only for the consensus sequences obtained by the resequencing approach, as we do not have a de novo assembled sequence for SpltNPV-G2b. Second, we consider mutations that are unique for each of the two isolates from Pakistan. These mutations can render information on the differences between the two isolates, including their different levels of virulence between these two isolates. This comparison can be performed for both the resequencing and de novo assembly analyses. In the results section, we will focus on the resequencing results as here they appear to be more conservative and detect fewer mutations.

### 3.5. dN/dS Analysis: Differences with the SpltNPV-G2b and Identification of Loci Associated with Virulence of the SpltNPV-Pak-Isolates

We performed dN/dS and dI/dS analyses to determine what types of selection have been acting on the whole genome and specific loci for the virus isolates we are investigating. As explained in Results Section 3.4, based on the similarity between the isolates we have considered (1) mutations that are shared by the two Pakistani SpltNPV isolates with respect to SpltNPV-G2b (henceforth “shared mutations”), and (2) mutations that are unique to each Pakistani isolate (henceforth “unique mutations”). If we consider the whole genome, for both the shared and unique mutations we find that purifying selection predominates (Table 2; see also Appendix A).

Only under a very high TTR (transition to transversion ratio) of 5 does the dI/dS for the unique mutations approach 1. Moreover, given that many noncoding mutations are being called in the HRs, we do not think this result is biologically meaningful. Therefore, our analysis clearly indicates that purifying selection has predominated at the level of whole genomes, when both shared and unique mutations for the SpltNPV isolates from Pakistan are considered.

Next, we performed dN/dS analysis of all individual genes, again dividing the analysis into shared and unique mutations. For the vast majority of genes, there was again purifying selection as dN/dS < 1 (Figure 2). However, we found values representative for neutral evolution or even directional selection for a small number of shared and unique mutations (Table 3). For the shared mutations, there were four genes with a dN/dS ≥ 1: ORFs 22, 65, 113, and 117. Functions of ORFs 22, 113, and 117 are unknown, although ORF22 does contain a chitin-binding domain. ORF65 codes for an RNA cap methyltransferase [22], and these mutations might therefore might have an effect on transcript stability or translation.

For the unique mutations, there were four genes with a dN/dS ≥ 1: ORFs 25, 41, 122, and 139. ORFs 25, 41, and 139 are short predicted ORFs with unknown functions, in which we found only a single nonsynonymous mutation. Therefore, we do not think these results are meaningful. However, ORF122 is larger and had three nonsynonymous mutations. Interestingly, if we consider the resequencing analysis we also note that all three mutations in ORF122 occurred in SpltNPV-Pak-TAX1, the fast-acting isolate, as compared to SpltNPV-Pak-BNG. ORF122 codes for a viral fibroblast growth factor (vFGF) protein [24], which assists in baculovirus escape from the midgut to the hemocoel prior to midgut cell sloughing off. Knockout of Helicoverpa armigera nucleopolyhedrovirus (HearNPV, group II alphabaculoviruses) *vfgf* resulted in a slower speed of kill of the virus [25], as do knockouts of its homolog in Group I alphabaculoviruses [24,26].

### 3.6. Limited Within-Isolate Polymorphism

Finally, we considered whether there was polymorphism within each Pakistani isolate by mapping reads against their respective de novo assembled genome. We chose to use these data because this approach is more conservative in calling variants than the resequencing approached used. In our experience, mutations identified as being fixed (i.e., “homozygous”) by the low frequency variant detector (see Materials and Methods) are rarely artifacts of poor mapping. However, mutations identified as being polymorphic (i.e., “heterozygous”) are sometimes mapping artefacts, and for all heterozygous variants called at a frequency > 0.02 we therefore manually curated the read mappings. This led to discarding most variants, typically because they appeared to be artifacts from the mapping of reads to regions with repeats (see also Appendix A). After this curation, we found surprisingly little polymorphism, and only SNPs (Table 4).

For SpltNPV-Pak-BNG, there were four SNPs at frequencies ranging between 0.021 and 0.470; for SpltNPV-Pak-TAX1 there was only a single SNP at a frequency of 0.139. The patterns in coverage for the mapping of reads against the reference genome (Appendix A) do not suggest that there is any copy number variation in the population. For both Pakistani isolates, there is therefore very limited within-isolate genetic variation, and the genotypes present must be very closely related, as they differ only by 1–4 nucleotides.

## 4. Discussion

We characterized and compared the SpltNPV-Pak-BNG isolate and the faster-acting isolate SpltNPV-Pak-TAX1 using a deep sequencing approach, and used two divergent approaches for data analysis: resequencing by mapping of reads to the SpltNPV reference sequence and de novo assembly. The results of these two approaches were similar, in terms of the presence or absence of large indels and the homozygous SNPs identified. This similarity highlights the power of a de novo assembly based approach despite the occurrence of repeat regions. However, we could only perform these analyses with longer-read MiSeq data, as de novo assembly failed for short read but higher coverage HiSeq data for SpltNPV-G2b, a derivative isolate from the SpltNPV type species G2.

The two Pakistani isolates showed approximately 99% nucleotide identity, as had been anticipated from sequencing of a small number of loci [4,5]. These two isolates were more distantly related to a Chinese isolate, SpltNPV-G2b. A dN/dS analysis showed that purifying selection predominates in the genome, i.e., nonsynonymous mutations have been fixed less often than expected by chance, suggested the amino acid sequence is being conserved. However, we were able to identify four genes under neutral or positive selection, suggesting that these genes might have played a role in adaptation as the virus radiated out to Pakistan. Unfortunately, the functions of most of these genes have not been identified, and we cannot make inferences on the selection pressures that have acted on the Pakistani SpltNPV populations.

There was limited genetic variation between the two Pakistani isolates. This variation included two indels, a partial deletion of ORF125 in SpltNPV-Pak-BNG and a complete deletion of homologous region hr17 in SpltNPV-Pak-TAX1, as well as approximately 250 nonsynonymous SNPs. Baculovirus homologous regions (hrs) are thought to be origins of DNA replication and transcriptional enhancers, though no single homologous region appears to be essential [27,28]. As hr17 might control transcription levels, it may also effect the virulence of the virus as a whole. When it serves as replication origin it may affect the replication speed and final outcome of the infection. dN/dS analysis suggests that ORF122 of SpltNPV-Pak-TAX1—coding for a vFGF—is not subject to strong purifying selection, and possibly even subject to directional selection. In other studies, both vFGFs and homologous regions have been linked to baculovirus virulence and speed of kill [24,25,26], making it plausible that these two loci contribute somehow to the increased speed of kill of the SpltNPV-Pak-TAX-1 isolate. Mutational analysis of ORF122 and hr17 using SpltNPV bacmids to be constructed or via CRISPR-CAS9 mutagenesis would be required to empirically test this claim. A different interpretation of these results is that the deletion of ORF125 (*broB*) contributes to the reduced virulence of SpltNPV-Pak-BNG, in which case we would expect to find this deletion in most of the Pakistani virus isolates. It is interesting to note that differences in biological activity were found between spatially separated isolates of Lymantria dispar multiple nucleopolyhedrovirus (LdMNPV) in conjunction with differences in whole genome sequence information [29,30]. By applying dN/dS analysis it is possible that other virulence genes come to light, an approach that has worked well when comparing different baculovirus species [31].

We found surprisingly little genetic variation within each Pakistani isolate, with only a single polymorphic SNP in BNG and four polymorphic SNPs within SpltNPV-Pak-TAX1. Although we had reasonably high sequencing coverage, in another study much higher coverage (>100,000) has been used to identify very rare variants [18]. Nevertheless, our results contrast with those generally found for other baculoviruses [15,17,18], where considerable genotypic variation is typically found. However, in the case of these two Pakistani SpltNPV isolates, there is very limited polymorphism. Although we cannot gauge the generality of this result given our sample size, it is still interesting to speculate on specific or general causes. The limited polymorphism observed could very well be due to a small number of viral founders of infection [16,32]. Alternatively, in another study the frequency of mixed-genotype infections was found to be lower during disease outbreaks [33], possibly due to the fixation of high virulence strains in epidemic conditions. A recent outbreak and subsequent dispersal of a SpltNPV-Pak ancestor also could help to explain limited variation in Pakistani isolates.

Regardless of how it has come about, the lack of polymorphism observed means that there is probably very little variation in speed-of-kill within these two isolates. Hence, using a laboratory evolution approach to select for variants with a faster speed-of-kill will be entirely dependent on the occurrence of de novo mutations and may therefore take considerable evolutionary time given the low mutation rates for DNA viruses. To use such an approach, it may be advisable to search for isolates with greater standing genetic variation, or alternatively, to mutagenize the virus population.

One important constraint in our study is the lack of data on mutational biases in insect viruses. Assuming different TTR (transition to transversion ratio) values clearly affects the results of the dN/dS analyses, but in the absence of any data, we could only perform the analysis for a range of values. For the dN/dS analysis on whole genomes, we found good evidence for strong purifying selection, regardless of assumptions on mutational bias. On the other hand, assumptions on the TTR clearly matter for the analysis of individual genes, with stronger evidence for directional selection when higher TTR values (>1) are assumed. These complications highlight the need for the investigation of the mutational spectrum [34] for insect DNA viruses. We used a simple counting method for the dN/dS analysis, but given the small number of mutations observed, and the fact that they affect different codons, this approach appears to be justified.

In conclusion, deep NGS sequencing of two Pakistani SpltNPV isolates with differences in speed of action revealed limited genetic variation within isolates. When comparing the Pakistani SpltNPV isolates, there was neutral to positive selection on one locus: ORF122 (*vfgf*). The fact that SpltNPV-Pak-TAX1 is more virulent than SpltNPV-Pak-BNG suggests this locus could be a determinant of virulence, as could hr17 that is absent in SpltNPV-Pak-TAX1. It should be worthwhile to deep-sequence other SpltNPV-Pak isolates with different speeds of action, such as SFD1, GRW1, to see whether the above observations have general validity. In addition, it would be of interest to sequence SpltNPV isolates outside of Pakistan, e.g., from China or Japan, to see the extent of variation within these isolates and which ORFs are subject to positive selection [35]. From a more practical point of view, these loci may serve as markers for selecting superior SpltNPV isolates. It would also be of interest to apply the dN/dS approach to other baculoviruses [36] to identify markers for improved performance of these viruses as biocontrol agents.

## Figures and Tables

**Figure 1 viruses-11-00872-f001:**
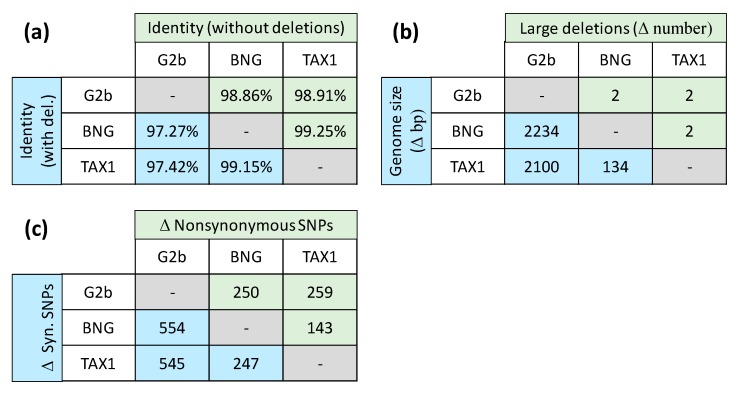
Pairwise comparisons are shown between the three resequencing-derived consensus sequences for SpltNPV-G2b (“G2b”), SpltNPV-Pak-BNG (“BNG”), and SpltNPV-Pak-TAX1 (“TAX1”). The positions above and below the diagonal contain different metrics, and these metrics are indicated to the far left and top of each table. Blue and green shading is used to indicate the pairwise comparison being shown in a panel. (**a**) The percentage shared identity without and with (“with del.”) large deletions included is given. (**b**) The difference in genome size and the difference in the number of large (>100 base pair (bp)) deletions is given. (**c**) The difference in the number of synonymous (syn.) and nonsynonymous single-nucleotide polymorphisms is given. Note that with the exception of the number of large deletions, SpltNPV-Pak-BNG and SpltNPV-Pak-TAX1 are always more similar to each other than to SpltNPV-G2b.

**Figure 2 viruses-11-00872-f002:**
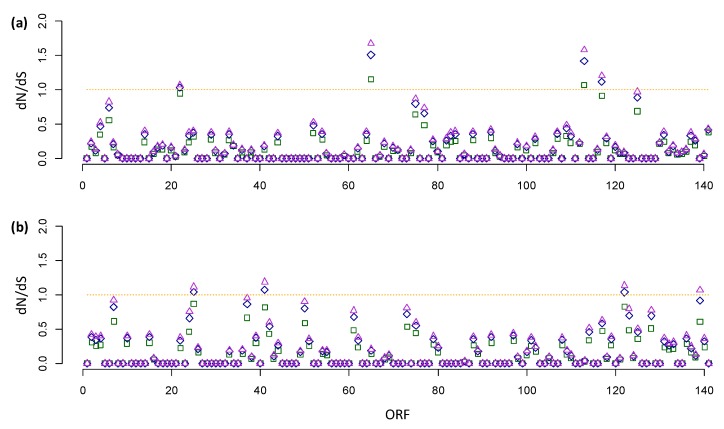
The ratio of the rate of nonsynonymous to the rate of synonymous substitution (dN/dS) analyses for fixed single nucleotide polymorphisms (SNPs), where in both panels the x-axis is the open reading frame (ORF) number corresponding to the SpltMNPV-G2b reference and the y-axis is the dN/dS value. dN/dS values assuming different mutational biases are presented: transition to transversion ratio (TTR) values of 1 (green squares), 3 (blue diamonds), and 5 (magenta triangles). The orange line indicates dN/dS = 1, which indicates neutral evolution. When dN/dS < 1, there is purifying selection, whereas when dN/dS > 1 there is positive selection. (**a**) Results for the analysis of shared mutations (mutations that are found in both Pakistani isolates but not in SpltNPV-G2b), based on the resequencing approach. (**b**) Results for the analysis of unique mutations found only in one of the Pakistani isolates. For details of the analysis for all ORFs with a result ≥ 1, see Table 3.

**Table 1 viruses-11-00872-t001:** Large deletions in the Splt-Pak-BNG and Splt-Pak-TAX1 isolates.

Isolate	Start Deletion ^a^	End Deletion ^a^	Size	Locus and Notes
SpltNPV-Pak-BNG	122,661	123,153	493	ORF125 (*broB*), partial deletion
	123,487	125,222	1736	ORF126 deleted
SpltNPV-Pak-TAX1	123,487	125,222	1736	ORF126 deleted
	137,718	138,035	319	hr17 deleted, 33 bp insert

^a^ Coordinates with respect to the SpltNPV-G2b reference sequence, indicating first and last missing nucleotide position. ORF stands for open reading frame.

**Table 2 viruses-11-00872-t002:** dN/dS and dI/dS analysis for whole genomes.

	Mutations		Obs. Mutations ^b^		dN/dS or dI/dS		
Analysis	Contrasted	Approach ^a^	NS/I	S	TTR ^c^ = 1	TTR ^c^ = 3	TTR ^c^ = 5
dN/dS	Shared	Resequencing	183	428	0.114	0.150	0.168
	Unique	Resequencing	125	247	0.135	0.178	0.199
		De novo	111	247	0.120	0.158	0.176
dI/dS	Shared	Resequencing	125	428	0.490	0.606	0.655
	Unique	Resequencing	110	247	0.748	0.924	1.000
		De novo	87	247	0.591	0.731	0.791

^a^ Approach refers to the next generation sequencing (NGS) data analysis. ^b^ The observed mutations in each class: NS = nonsynonymous, I = intergenic, and S = synonymous. Note NS or I is given depending on the analysis. ^c^ The transition to transversion ratio (TTR) assumed in the ratio of the rate of nonsynonymous to the rate of synonymous substitution (dN/dS) or the rate of intergenic to synonymous substitution (dI/dS) analysis.

**Table 3 viruses-11-00872-t003:** Genes under neutral or purifying selection for the dN/dS analysis.

			Mutations ^c^		dN/dS			
Mutations	ORF ^a^	AA ^b^	NS	S	TTR ^d^ = 1	TTRd ^d^ = 3	TTR ^d^ = 5	Function
Shared	22	114	5	0	0.947	1.027	1.065	Unknown, contains ChtBD2 chitin-binding domain
	65	313	4	1	1.150	1.505	1.672	Putative RNA cap (nucleoside-2’-O)-methyltransferase, AcMNPV orthologue: *ac69*) ^e^
	113	401	4	1	1.064	1.413	1.578	Unknown, gene found in many baculoviruses (AcMNPV orthologue: *ac18*), not essential ^f^.
	117	105	3	1	0.910	1.112	1.206	Unknown
Unique	25	59	1	0	0.872	1.040	1.117	Unknown, only found in SpltNPV
	41	54	1	0	0.816	1.069	1.188	Unknown, only found in SpltNPV and SpliNPV
	122	246	3	1	0.824	1.036	1.134	Viral fibroblast growth factor (vFGF), determinant of speed of kill ^g^
	139	60	1	0	0.608	0.918	1.069	Unknown

^a^ Only open reading frames (ORFs) with a the ratio of the rate of nonsynonymous to the rate of synonymous substitution (dN/dS) value that exceeds 0.95 for any transition to transversion ratio (TTR) value are included here. Open reading frame (ORF) 125 has been excluded here, because the resequencing analysis calls spurious mutations in the regions flanking the large genomic mutation. ^b^ The length of the ORF in amino acids. ^c^ The observed mutations in each class: NS = nonsynonymous and S = synonymous. ^d^ The TTR assumed in the dN/dS analysis. ^e^ See [22]. ^f^ See [23]. ^g^ See [24,25,26].

**Table 4 viruses-11-00872-t004:** Within-isolate polymorphism.

Isolate ^a^	Mutation	Count/Coverage	Frequency	Forward/Reverse	ORF	AA Change
BNG	c35,000a	155/3881	0.0399	44.19%	ORF36	none
	c35,202t	1623/3455	0.4698	49.97%	ORF36	none
	g41180a	60/2910	0.0206	47.76%	ORF42	none
	c111904t	407/3822	0.1065	49.34%	ORF116	S45F
TAX1	c98204a	416/2995	0.1389	43.56%	none	none

^a^ The SpltNPV isolate: “BNG” for SpltNPV-Pak-BNG and “TAX1” for SpltNPV-Pak-TAX1. “None” indicates no open reading frames (ORFs) or amino acid (AA) sequences are changed.

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
