# Peer review of "Identification of Loci Associated with Enhanced Virulence in Spodoptera litura Nucleopolyhedrovirus Isolates Using Deep Sequencing"

_viruses, 2019, doi:10.3390/v11090872_

Round 1

Reviewer 1 Report

This manuscript describes the comparison of genomic sequences of two Spodoptera litura nucleopolyhedroviruss (SpltNPVs) isolated in Pakistan and identification of loci and genes involved in its enhanced virulence. This study is interesting to researchers and can be evaluated as a paper in “Viruses”. However, this manuscript still has some doubts. Some comments are described below.

In this manuscript, “Spodoptera litura” and “Spodptera littoralis” should be italic.

The authors use “homozygous SNPs”. In this study, this word, “homozygous SNPs”, is not appropriate and the authors should use other word.

Line 436, ORF7 is shown first. What is “ORF7”? Why did the authors pick this “ORF7” up?

In Figure S3, ORF4 is strayed from the line with ORF125. What “ORF4” is and why “ORF4” is strayed from the line is not mentioned.

SpltNPV-Pak-BNG has partially deleted ORF125, but SpltNPV-Pak-TAX1 does not have. Is it possible that the deletion of ORF125 is also involved in the virulence?

Author Response

We thank the reviewers for their constructive and helpful comments that have helped improve the manuscript. Below, we give the reviewer comments in italics and our responses in roman text. Line numbers refer to the track changes version of the manuscript for easy reference.

This manuscript describes the comparison of genomic sequences of two Spodoptera litura nucleopolyhedroviruss (SpltNPVs) isolated in Pakistan and identification of loci and genes involved in its enhanced virulence. This study is interesting to researchers and can be evaluated as a paper in “Viruses”. However, this manuscript still has some doubts. Some comments are described below.

We thank the reviewer for a positive outlook on our study, and have taken onboard these useful suggestions.

In this manuscript, “Spodoptera litura” and “Spodptera littoralis” should be italic.

We thank the reviewer for alerting us to inconsistencies, and have double-checked our manuscript for the incorrect use of italics or other mistakes in species names and made corrections were necessary (title and Line 238). According to the ICTV guidelines, a virus name (i.e., referring to physical entities and not a taxonomic classification) should not be in italics “even when it includes the name of a host species or genus (https://talk.ictvonline.org/information/w/faq/386/how-to-writevirus-and-species-names). Therefore, in these instances full virus names should have the insect species name in roman text. By contrast, we refer to a virus species (i.e., taxonomic classification) once, and have italicized the full virus name in this case (Lines 91-93) as per the ICTV guidelines.

The authors use “homozygous SNPs”. In this study, this word, “homozygous SNPs”, is not appropriate and the authors should use other word.

We thank the reviewer for pointing out this error, and now use the term “fixed” instead of “homozygous” to qualify SNP where appropriate. We do use “homozygous” when describing the results of low frequency variant detector, although by using quotation marks we make clear that this refers to software outputs (Lines 418-422).

Line 436, ORF7 is shown first. What is “ORF7”? Why did the authors pick this “ORF7” up?

We thank the reviewer for being scrupulous and pointing out this discrepancy. In a first analysis based only on the resequencing approach (which appears to be less conservative than the de novo assembly in identifying mutations), we also had a hit for ORF7 in the dN/dS analysis. The conclusion paragraph has been amended to reflect the improved final version of our analysis (Lines 500-511).

In Figure S3, ORF4 is strayed from the line with ORF125. What “ORF4” is and why “ORF4” is strayed from the line is not mentioned.

For both ORFs 4 and 125 the resequencing approach identifies more mutations than the de novo assembly approach. In the specific case of ORF4, the hoar gene, resequencing identifies more mutations because there are clearly complications with the alignment. We point this out in the results section (Lines 273-278).

SpltNPV-Pak-BNG has partially deleted ORF125, but SpltNPV-Pak-TAX1 does not have. Is it possible that the deletion of ORF125 is also involved in the virulence?

We think it is indeed possible that a partial deletion of ORF125 is involved in lower virulence of SpltNPV-Pak-BNG. SpltNPV-Pak-TAX1 has a faster speed of kill then most Pakistani isolates (including SpltNPV-Pak-BNG) (see Reference 5), so we think it is more likely that the changes that affect virulence will be found in SpltNPV-Pak-TAX1. Nevertheless, without having genome sequences for more isolates with different levels of virulence, we cannot be sure. We now also mention this alternative interpretation in the conclusion (lines 463-466).

Reviewer 2 Report

This paper compared two isolates of SpltNPV genomes, which have different virulence to the host. The authors found that the absence of hr17 in TAX1 isolate, and the absence of ORF125 in BNG isolate. And the positive selection in ORF122. In the final part of this abstract, the author further mentioned that they identified four ORFs under positive selection that might be related to the general adaptation of SpltNPV to Central Asia. Moreover, two loci were also identified as possibly linked to the enhanced virulence of the TAX1 isolate. As I listed above, in the abstract, it is hard to quickly understand the main finding of this research. Which one is the key gene involved in the virulence? hr17 ORF125 ORF122 or four genes? or two loci? The author should make it clear in the abstract.

In the paper, the major work is sequencing and comparison of two genetically identical SpltNPV isolates. The author found some results of interesting, however, this data doesn’t reach the quantity of this journal yet. Otherwise, the functional analysis of these potential genes or loci could be done to get solid data to demonstrate their final hypothesis. 

Author Response

We thank the reviewers for their constructive and helpful comments that have helped improve the manuscript. Below, we give the reviewer comments in italics and our responses in roman text. Line numbers refer to the track changes version of the manuscript for easy reference.

This paper compared two isolates of SpltNPV genomes, which have different virulence to the host. The authors found that the absence of hr17 in TAX1 isolate, and the absence of ORF125 in BNG isolate. And the positive selection in ORF122. In the final part of this abstract, the author further mentioned that they identified four ORFs under positive selection that might be related to the general adaptation of SpltNPV to Central Asia. Moreover, two loci were also identified as possibly linked to the enhanced virulence of the TAX1 isolate. As I listed above, in the abstract, it is hard to quickly understand the main finding of this research. Which one is the key gene involved in the virulence? hr17 ORF125 ORF122 or four genes? or two loci? The author should make it clear in the abstract.

We thank the reviewer for suggesting the main message of the paper must be clearer in the abstract.
We have rewritten the abstract to better convey the main message of the paper (Lines 25-43).

In the paper, the major work is sequencing and comparison of two genetically identical SpltNPV isolates. The author found some results of interesting, however, this data doesn’t reach the quantity of this journal yet. Otherwise, the functional analysis of these potential genes or loci could be done to get solid data to demonstrate their final hypothesis.

On this point we respectfully disagree with the reviewer. We are convinced our work is an important scientific advance that will be appreciated by the community, including insect virologists, biocontrol specialists, and bioinformaticians. First, we do not think this comment is a fair reflection of our work, as this manuscript represents a major effort and an appreciable volume of highly relevant results. The focus is on viral genomics and bioinformatics analyses, so we find it unfair to dismiss work because follow-up experimental tests have not been performed yet. Second, there is very little work and information on the genetic basis of viral traits that are desirable for biological control purposes. So, there is an urgent need to have a ‘quick scan’ (as we promote here) to select potential candidates and reduce the need of bioassaying cumbersomely many baculovirus isolates. Here, for a non-modelsystem virus that has potential for biological control purposes, we show that genomics approaches can be useful for making inferences on genes of interest. We think that the current paper is a groundbreaking and stimulating contribution, not necessarily because we identify new loci linked to speed of kill, but because we give a “proof or principle” that this approach works. Third, although our study is focused on sequencing and bioinformatics, we have been very thorough in performing and validating these analyses. We think that our work therefore also makes a methodological contribution to how such analyses can be done. Finally, it should be kept in mind that we are not working with a model system. While we absolutely agree on the value of further experimentation to test the hypotheses we have put forward, that is a major project beyond the intention and scope of the current study. The experiments the reviewer has suggested would require tools that do not exist at present, such as bacmids and cell culture. We have outlined this outlook in the discussion of this
paper.

Round 2

Reviewer 2 Report

The authors have explained and defence for their manuscript, I therefore have no comment.